# Practical Aspects of Upper Gastrointestinal Bleeding in Children

**DOI:** 10.3390/jcm12082921

**Published:** 2023-04-17

**Authors:** Lucia Maria Sur, Ionel Armat, Genel Sur, Ioana Badiu Tisa, Madalina Adriana Bordea, Iulia Lupan, Gabriel Samasca, Calin Lazar

**Affiliations:** 1Department of Pediatrics I, Iuliu Hatieganu University of Medicine and Pharmacy, 400012 Cluj-Napoca, Romania; 2Department of Pediatrics II, Iuliu Hatieganu University of Medicine and Pharmacy, 400012 Cluj-Napoca, Romania; 3Department of Pediatrics III, Iuliu Hatieganu University of Medicine and Pharmacy, 400012 Cluj-Napoca, Romania; 4Department of Microbiology, Iuliu Hatieganu University of Medicine and Pharmacy, 400012 Cluj-Napoca, Romania; 5Department of Molecular Biology, Babes Bolyai University, 400084 Cluj-Napoca, Romania; 6Department of Immunology, Iuliu Hatieganu University of Medicine and Pharmacy, 400012 Cluj-Napoca, Romania

**Keywords:** upper gastrointestinal bleeding, actualities, approach, child

## Abstract

Upper gastrointestinal bleeding (UGB) in children is a potentially life-threatening condition that represents a challenge for pediatricians and pediatric surgeons. It is defined as bleeding from any location within the upper esophagus to the ligament of Treitz. UGB can have many causes that vary with age. The impact on the child is often proportional to the amount of blood lost. This can range from mild bleeding that is unlikely to cause hemodynamic instability, to massive bleeding that requires admission to the intensive care unit. Proper and prompt management are very important factors in reducing morbidity and mortality. This article aims to summarize current research regarding the diagnosis and treatment of UGB. Most of the data used in the literature published on this subject is extrapolated from adulthood.

## 1. Introduction

Upper gastrointestinal bleeding (UGB) is a medical emergency caused by intraluminal bleeding within the gastrointestinal tract from the esophagus to the ligament of Treitz. In recent years, UGB management has changed significantly due to improved guidelines, correct resuscitation, and improved endoscopic diagnosis and treatment techniques. UGB clinical manifestations vary from self-limited bleeding to life-threatening situations. In minor bleeding cases, there is no need for therapeutic intervention. Children with major bleeding necessitate a synergetic approach to assess hemodynamic stability, estimate blood loss, and determine the etiology and origin of the bleeding [1]. Clinical manifestations of a massive UGB in children include cold extremities, thirst, delayed capillary refill time, pallor, delirium, confusion, dizziness, and coma. UGB has many causes that can be categorized by age group. Other causes include vitamin K deficiency, stress gastritis, coagulopathy, duplication cysts, and trauma from placing a nasogastric tube [2]. In infants, common causes of UGB are gastritis, gastro-duodenal ulcers, cirrhosis with esophageal varices, and foreign body ingestion. In older children and adolescents, the main causes are Mallory-Weiss syndrome, Dieulafoy’s lesions, and esophageal and gastric varices. The most frequent cause of UGB found in children were erosive gastritis (33.5%), followed by duodenal ulcer (23.2%), gastric ulcer (9%), erosive duodenitis (3.8%), erosive esophagitis (3.5%), varices (2.4%), and Mallory-Weiss tears (0.8%) [3]. It is very difficult to determine the worldwide incidence and prevalence of UGB. There was reported an incidence of 1 to 2 per 10,000 children per year [4]. The mortality rate for UGB in children can vary from 5% to 15%, depending on the etiology and medical care available in the region. In the PICU (Pediatric Intensive Care Unit) the risk of UGB is higher, ranging from 16–25% [5].

This article aims to summarize current research regarding the diagnosis and treatment of UGB. We searched the PubMed database for articles with the keywords “upper gastrointestinal bleeding”. We included in our research only articles with references to children. We focused on the most important medical aspects of upper gastrointestinal bleeding: What are the clinical features? How do we make a clinical diagnosis? How do we manage this disease? We also focused on transcatheter arterial embolization for refractory upper gastrointestinal bleeding in children because our research indicates that this method merits more clinical attention in the future as a minimally invasive solution for massive hemorrhages in hemodynamically unstable patients.

## 2. Clinical Features

Gastrointestinal bleeding can be classified as major if there is a decrease in serum hemoglobin of more than 2 g/dL and/or a need for a blood transfusion. If the bleeding does not cause hemodynamic instability, it is classified as minor [6].

The most common signs of UGB are hematemesis and melena. There are other signs and symptoms less specific than these, such as abdominal pain, dizziness, and hypovolemic shock. A population-based survey in France conducted by Lamiae Grimaldi-Bensouda included 177 children with UGB. Among these, 96.6% presented hematemesis, 14.1% melena, and 2.8% hypovolemic shock [1]. A similar study in China included 1218 children and showed the following results regarding clinical presentation: hematemesis 59.3%, melena 22.6%, hypovolemic shock 2.2%, abdominal pain 46.2%, and dizziness 6.9% [4]. We present the medical causes of UGB organized by group age in Table 1.

An important cause of UBG is pill esophagitis. Some prescribed drugs induce esophageal lesions, causing direct esophageal mucosal injury or systemic effects. More than 100 different medications have been reported to cause pill esophagitis [7]. The most common medications associated with UGB are NSAIDs, specific antibiotics, and bisphosphonates. Tetracyclines, which are normally used to treat acne in teens, are the most common antibiotic class implicated in pill esophagitis [8]. Other antibiotics associated with medication-induced esophagitis are penicillin, clindamycin, lincomycin, sulfamethoxypyridazine, and azithromycin [9].

## 3. Diagnosis

The onset of signs and symptoms of UGB is worrying for parents and children, which is why they come to the emergency room as soon as possible. Accurate and efficient diagnosis is a challenge for pediatric practitioners because early treatment provides better outcomes and reduces mortality. Proper diagnosis protocol includes medical history, physical examination, blood work, and diagnostic procedures. We present a brief overview of the diagnostic steps for UGB in Figure 1.

### 3.1. History

A first-look assessment is important to establish if a life-threatening situation is at hand. In emergencies, if the patient is hemodynamically unstable, the medical history portion of the diagnosis must be very brief. Patients with major current illnesses such as respiratory failure, cardiac failure, or sepsis may present with stress ulcers or stress gastritis. Variceal bleeding may be suspected in children with cirrhosis, cystic fibrosis, biliary atresia, portal vein thrombosis, or Budd-Chiari syndrome. In neonates, it is important to determine if a vitamin K dose has been delivered. Furthermore, family history is important. In toddlers, the corrosive ingestion of various household substances must be taken into account. In this age group, ingestion of foreign objects is common. The most dangerous are button batteries [2,5].

### 3.2. Physical Examination

In all UGB situations, the clinical examination should begin with the airway, breathing, and circulatory system to determine whether or not the patient shows signs of is hemodynamic instability. The vital signs must be monitored. It is known that tachycardia is the most sensitive indicator of blood loss in children [8]. Conducting an abdominal examination is important to rule out surgical emergencies. Bruises may suggest trauma or bleeding disorders. Rectal examination is also important in patients with UGB [1].

### 3.3. Laboratory

One of the primary objectives of initial laboratory tests is to determine the patient’s blood type and cross-match if a transfusion is needed. Hemoglobin (Hb) and hematocrit determination are part of the standard procedure. Initial Hb may be normal [2]. Platelet count and coagulation factors may indicate that a platelet transfusion or cryoprecipitate is required. High blood urea means that either a significant bleed took place or that there is insidious bleeding. Conducting a serum creatinine test is also a routine investigative procedure. If the creatinine level is high and correlates with low Hb and high urea, hemolytic uremic syndrome is plausible.

### 3.4. Diagnostic Procedures

Confirmation of a UGB can be made by placing a nasogastric or orogastric tube if there is a return of blood or coffee grounds. This procedure is important for preparing patients for gastrointestinal endoscopy and is also useful for therapeutic purposes. Upper gastrointestinal endoscopy is the most useful procedure for the diagnosis and treatment of UGB. Performing an endoscopy is essential in diagnosing and determining the etiology of UGB. Endoscopic diagnostic procedures are safe for children. A large survey of 2046 pediatric upper gastrointestinal endoscopies reported only two major complications and no deaths [10]. One limitation of the endoscopy is the need for general anesthesia and intubation.

### 3.5. Radiological Diagnosis

For this article, we will discuss only CT angiography (CTA) and nuclear RBC scans. In recent years, CTA has become one of the most commonly used radiological diagnostic methods for gastrointestinal bleeding. With a bleeding detection threshold of 0.3–0.5 mL/min, CTA is similar to scintigraphy with tagged red blood cells and superior to angiography [11]. This method can locate the source of bleeding, differentiate venous bleeding from arterial bleeding, and is usually available in emergency settings. In UGB this method is considered a second-line choice in patients with negative endoscopy or in cases where the bleeding source was not identified during an endoscopy [12].

A nuclear RBC scan is a noninvasive diagnostic test used to evaluate patients with suspected gastrointestinal bleeding. It involves injecting small amounts of radioactively marked red blood cells (RBC), usually 99 mTc-RBCs. This radionuclide imaging method is used to determine the status of bleeding and its location and can approximate the bleeding volume [13]. The detection rate is accurate at levels as low as 0.1 mL/m [14].

## 4. Management

After the initial diagnosis, it is crucial to determine the risk group in which the child will be placed. Is it a massive hemorrhage associated with hemodynamic instability or a mild, stable digestive hemorrhage?

### 4.1. Management of the Massive UGB

According to University Hospitals of Leicester guidelines, high-risk patients are the following: Patients in shock (hypotension, confusion, dizziness, tachycardia, acidosis, prolonged capillary refill time), patients with active ongoing bleeding, patients with documented esophageal varices, liver failure, renal failure, heart failure or congenital heart disease, different types of coagulopathies, and also Hb < 8 mg/dL [1].

All hemodynamically unstable patients should be managed according to PALS (Pediatric Advanced Live Support).

Airway: PALS guidelines from 2021 recommend starting oxygen therapy at SpO_2_ <94%. In patients known to have chronic conditions, oxygen is recommended if SpO_2_ is 3% below the baseline. In the presence of any signs of aspiration pneumonia or in conditions where oxygen saturation cannot be maintained above 90, intubation is recommended [2].

Breathing: As a consequence of blood loss, tachypnea is a normal adaptive response. The goal is to keep SpO_2_ between 94 and 98% with as little supplemental oxygen as possible.

Circulation: in severe cases, intraosseous access may be necessary. The goal is to have a minimum of two large bore cannulas. If there is a need for inotropic treatment, central vein access is recommended. A blood transfusion is recommended and should be performed as soon as possible. It is also important to correct the platelet count if necessary. The University Hospitals of Leicester guideline for UGB recommends the following ratio for blood transfusion: 4 units of blood: 3 units of fresh frozen plasma (FFP) and 1 unit of platelets. It is important to avoid having high systolic blood pressure.

### 4.2. Drugs

In the management of massive UGB, the following drugs are of great importance both in stopping bleeding and in supporting the patient in hemorrhagic shock:Octreotide is an effective treatment for variceal UGB [9]. The loading dose for intravenous infusion is 1 mcg/kg (maximum 50 mcg); the maintenance dose is 1–3 mcg/kg/h.Tranexamic acid is effective in reducing mortality in UGB and should be considered in all age groups [15]. In UGB, the recommended dose is 15 mg/kg (maximum 1 g).Phytomenadione (Vitamin K), 300 micrograms/kg IV (maximum 10 mg).Inotropic drugs should be considered in hemorrhagic shock with hypotension.Esomeprazole IV therapy decreases the recurrent bleeding rate [16]. Pediatric dose: <20 kg, 10 mg/day; >20 kg, 20 mg/day; 12–17 years, 40 mg/day.

### 4.3. Endoscopy Management

In all cases of massive gastrointestinal bleeding, an early endoscopy should be performed after initial resuscitation and hemodynamic stabilization. In cases of variceal bleeding, the endoscopy is recommended to be done in the first 12 h after admission [6,17]. In non-variceal bleeding, the recommended time for endoscopy is within 24 h after admission [18]. Most of the time, in pediatric patients, UGB has a non-variceal etiology. Although upper gastrointestinal endoscopy is the first line of treatment in UGB, it has the following limitations: an anesthesiologist is needed for intubation and sedation; for infants and young children, the devices are not small enough for all therapeutic procedures.

In cases of non-variceal bleeding, there are various methods to stop the bleeding. Injection of vasoactive substances is a commonly used method, usually 1 in 10,000 strength adrenaline or simple saline. Adrenalin injection produces short-term hemostasis and should always be used in combination with other methods (mechanical or thermocoagulation) [6]. Cautery techniques include electrocautery probes, heater probes, and argon plasma coagulation. Using electricity, heat, or argon ionized gas, these methods stop the bleeding by coagulating the blood vessels. Laser therapy is also a useful technique to stop non-variceal bleeding, but it is not used frequently due to its high costs. Mechanical methods are very effective for achieving hemostasis. This method includes band ligation and clip applications. Endoscopic clips are also known as hemoclips and are used in different pathologies, including Mallory Weiss tears and Dieulafoy’s lesions. After being applied, a hemoclip remains in place for a few days or weeks [19].

In UGB with variceal etiology, the most common endoscopic hemostatic procedures are sclerosing therapy and variceal ligation or banding. In association with adrenaline, sclerosing agents such as polidocanol, sodium morrhuate, ethanolamine, or sodium tetradecyl can be used to stop an acute esophageal variceal bleed. At present, variceal banding is the therapy of choice for variceal upper gastrointestinal bleeding because it is more effective compared to sclerotherapy at achieving lasting hemostasis [20].

### 4.4. Angiographic Embolization

If endoscopic procedures fail to control the bleeding, transcatheter arterial embolization is a fast, safe, and minimally invasive alternative [21]. Angiographic interventions have shifted from only having a diagnostic role to becoming a major therapeutic option. Usually, the source of the bleeding is determined by endoscopy, which is extremely beneficial to the angiographer. There are two main methods to control the bleeding: infusion of a vasoconstricting medication and mechanical occlusion [22]. Considering the progress made in pediatric interventional cardiology and the development of necessary materials for closing small blood vessels, these methods can be life-saving procedures in the case of a massive hemorrhage that cannot be controlled endoscopically [23,24,25,26,27].

### 4.5. Transjugular Intrahepatic Portosystemic Shunt (TIPS)

A transjugular intrahepatic portosystemic shunt is a well-known procedure used for portal hypertension complications. All conditions that interfere with blood flow at any level of the portal system can lead to portal hypertension [28]. The underlying cause of portal hypertension in children may be extrahepatic, like extrahepatic portal vein thrombosis, or intrahepatic, like cryptogenic cirrhosis, biliary atresia, Wilson’s disease, congenital hepatic fibrosis, and autoimmune hepatitis [29]. Indications for TIPS in children include relapsing bleeding or relapsing large varices after endoscopic treatment [29]. A significant number of pediatric patients after TIPS procedures showed the disappearance of varices at endoscopic follow-up [30].

## 5. Conclusions

UGB represents a serious, life-threatening condition that requires prompt diagnosis and therapeutic management. History and clinical exams help in identifying the cause. Endoscopy is the most valuable resource for diagnosis and treatment, but the main risks of upper gastrointestinal endoscopy are bleeding, aspiration, infection, and perforation, as well as anesthesia complications. Therefore, resuscitation and early endoscopy should be performed following specific guidelines. There are very few resources in the literature regarding arterial embolization in children.

## 6. Future Directions

One of the possible prominent endeavors of UGB diagnosis and treatment could be the application of machine learning (ML) techniques. ML has found its fingerprint in the advancement of several fields, including soil rehabilitation, the optimization of membrane fabrication for water decontamination, biofuel production, and nanomaterial characterization. The application of ML in medical diagnosis and treatment could be the inflection point required to boost scientific advancement in this field.

## Figures and Tables

**Figure 1 jcm-12-02921-f001:**
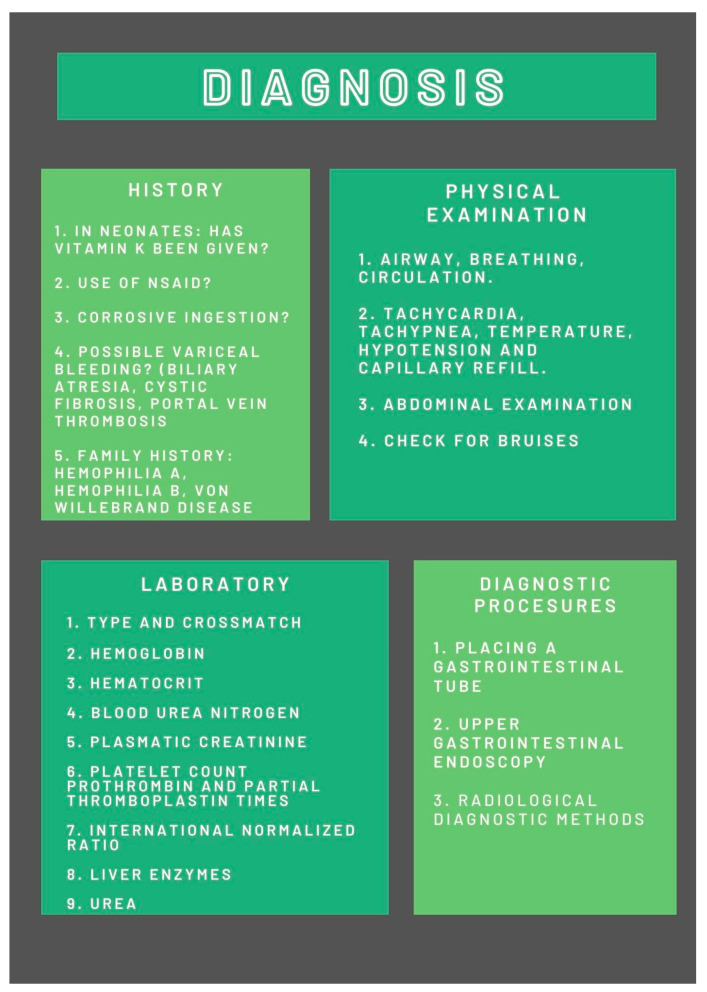
Diagnosis of UGB.

**Table 1 jcm-12-02921-t001:** The medical causes of UGB as per age group.

Causes of UGB as Per Age Group			
Neonates	1–6 Months	6 Months–5 Years	5 Years and Older
Swallowed maternal blood *	Cow’s milk protein allergy	Mallory Weiss tears	Gastritis
Vitamin K deficiency	Esophagitis	Dieulafoy’s lesions	Esophagitis
Stress gastritis/sepsis	Gastritis	Esophageal varices	Peptic Ulcer
Trauma (Nasogastric tubes)	NSAID-induced ulcer	Gastric varices	Esophageal varices
Duplication cyst	Esophageal varices	Foreign body ingestion	Gastric varices
Necrotising Enterocolitis	Gastric varices	NSAID-induced ulcer	Crohn’s disease (rare)

* Swallowed maternal blood is not the cause of UGB but has the same clinical manifestations.

## Data Availability

Not applicable.

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
