# Peer review of "Practical Aspects of Upper Gastrointestinal Bleeding in Children"

_jcm, 2023, doi:10.3390/jcm12082921_

Round 1

Reviewer 1 Report

Nice review of the upper gastrointestinal bleed in children.

You can discuss more about the diagnostic investigations like CT angiography and Nuclear RBC scans.

line 195 - Source of bleeding can be determined radiological tests also.

Angiographic embolization is not useful for gastric and esophageal varices, you need TIPS ( transjugular intrahepatic portosystemic shunt) and BRTO ( balloon-occluded retrograde transvenous obliteration) procedures.

Transhepatic embolization of varices can also be done in selected cases.

Author Response

I appreciate your comments and suggestions made. I added two new
paragraphs where I discuss more CT angiography and Nuclear scans. I also
write about TIPS in childrens along with relevant references.
I hope that the modified form of the manuscript now corresponds to your
requirements
Thank you

Reviewer 2 Report

In line 41 and in table 1, the authors claim that the swallowing of maternal blood is a cause of upper gastrointestinal bleeding in newborns: I think this concept should be clarified (is it a cause of bleeding or just a cause of hematemesis/melena, without active bleeding?).

The sub-paragraph "3.4: Diagnostic procedures" could be improved. 

In the paragraph "6. Future directions" the author focus the reader's attention on the machine learning techniques without any actual in-depth application or suggestion on the specific topic.

Author Response

I appreciate your comments. I made some changes in the manuscript text and
new bibliography was added. I corrected the appointed statement about
gastrointestinal bleeding in newborns. Regarding diagnostic procedures I
added two new paragraphs where I discuss more CT angiography and
Nuclear scans.
I hope that the modified form of the manuscript now corresponds to your
requirements
Thank you!

Round 2

Reviewer 1 Report

Nice review of upper GI bleed in children